# Albumin in patients with liver disease shows an altered conformation

Margret Paar[1,6], Vera H. Fengler[1,6], Daniel J. Rosenberg[2,3], Angelika Krebs[4], Rudolf E. Stauber [5], Karl Oettl [1,7✉] & Michal Hammel [2,7✉]

Human serum albumin (HSA) constitutes the primary transporter of fatty acids, bilirubin, and other plasma compounds. The binding, transport, and release of its cargos strongly depend on albumin conformation, which is affected by bound ligands induced by physiological and pathological conditions. HSA is both highly oxidized and heavily loaded with fatty acids and bilirubin in chronic liver disease. By employing small-angle X-ray scattering we show that HSA from the plasma of chronic liver disease patients undergoes a distinct opening compared to healthy donors. The extent of HSA opening correlates with clinically relevant variables, such as the model of end-stage liver disease score, bilirubin, and fatty acid levels. Although the mild oxidation of HSA in vitro does not alter overall structure, the alteration of patients' HSA correlates with its redox state. This study connects clinical data with structural visualization of albumin dynamicity in solution and underlines the functional importance of albumin's inherent flexibility.

[1] Division of Physiological Chemistry, Otto-Loewi Research Center, Medical University of Graz, Graz, Austria. [2] Molecular Biophysics and Integrated Bioimaging, Lawrence Berkeley National Laboratory, Berkeley, CA, USA. [3] Graduate Group in Biophysics, University of California, Berkeley, CA, USA. [4] Science Technology Interface–Structural Biology, Center for Medical Research, Medical University of Graz, Graz, Austria. [5] Department of Internal Medicine, Medical University of Graz, Graz, Austria. [6] These authors contributed equally: Margret Paar, Vera H. Fengler. [7] These authors jointly supervised this work: Karl Oettl, Michal Hammel. ✉email: karl.oettl@medunigraz.at; mhammel@lbl.gov

Human serum albumin (HSA) is a 66.5 kDa protein and is the most abundant protein in human plasma with a concentration of approximately 35–52 g/L. One of the most important physiological functions of albumin is the transport of endogenous and exogenous compounds. This involves binding the compound, the transport in plasma, and the compound's release at its target tissue. Several attempts have been made to find a test for the transport function and a function-structure-relation[1,2]. In in vitro studies, equilibrium dialysis is often used to investigate the binding of small molecules to HSA[3], while ultrafiltration techniques and titration curves are used to determine binding and dissociation constants of compounds[4,5]. The binding of spin-labeled fatty acids (FA) at different concentrations and with the addition of various amounts of ethanol is used to determine the detoxification and binding efficiency of albumin[2]. However, the detoxification efficiency of HSA may vary in vivo due to the conformational changes upon interactions with surfaces, like cells[6,7], or upon binding of different ligands, foremost FA and bilirubin[8,9]. This is especially relevant in liver disease as both FA and bilirubin in plasma may be increased substantially in these pathologies. HSA has been widely studied with analytical methods to characterize the most abundant modifications affecting HSA structure relevant to liver diseases[10]. Consequently, in medicine and clinical issues, HSA represents a potent biomarker. It was shown that the redox state of HSA could serve as a prognostic marker for survival in patients with end-stage liver diseases[11]. The redox state of HSA represents a prognostic marker. It influences its binding capacity, which seems crucial as its most essential functions are FA transport[12], drug binding and transport, and metal chelation[11,13,14].

Within aging and end-stage liver diseases, the redox state of HSA is shifted to the oxidized fractions, with implications for its transport function[11,14–16]. HSA can be separated into three different fractions according to the redox state of its cysteine residue 34 (Cys-34): (i) human mercaptalbumin (HMA), (ii) human non-mercaptalbumin-1 (HNA1), and (iii) irreversibly oxidized human non-mercaptalbumin-2 (HNA2)[17,18]. The redox state's impact on the HSA conformation and dynamicity is unknown and was further investigated here.

In vitro, the binding affinity of HSA is measured with defatted HSA. However, under normal physiologic conditions, HSA binds with approximately 0.1–2 moles of FA per mole of protein[19]. Several studies report that FA may influence the HSA-binding affinity[20–24]. HSA X-ray crystal structures have been studied concerning structural differences between unliganded HSA and HSA–myristate (HSA–MYR) complex[8,25]. These crystal structures suggested opening the HSA domains I and III relatively to the central domain II upon lipid binding. These studies brought the first light into the conformational changes that HSA may undergo in the human bloodstream upon lipid binding. Crystal static structures provide the foundation for the definition and interpretation of functional movements crucial to mechanistic understanding impacting HSA detoxification efficiency. Therefore, we herein leverage HSA crystal structures to interpret small-angle X-ray scattering (SAXS) data and determine conformational changes of HSA purified from human donors. We show that the HSA dynamicity in solution correlates with the increased level of lipid, bilirubin, and redox state and informs the structural mechanism relevant to liver disease. SAXS, which measures thermodynamic solution-state outside of crystal solid-state conditions[26], unveils the dramatic differences in the inherent flexibility of HSA domains I and III between healthy and patient donors. The distinct opening of HSA domains I and III in HSA purified from patients with chronic liver failure correlates with FA and bilirubin content, as well as the redox state of HSA. To the best of our knowledge, for the first time, our study connects clinical data with structural visualization of conformational changes of HSA.

## Results

**The biochemical diversity of patients' HSA**. In this study, we isolated and analyzed the HSA of 8 patients (2 females and 6 males) aged 42–81 years with chronic liver failure and of 4 healthy control persons (2 women and 2 men) aged 23–55 years by affinity chromatography. Thereby, HSA was separated from plasma ingredients and unbound ligands. Commercial HSA (Sigma HSA) served as another control.

The relation of FA per mole HSA ranged from 0.6 to 3 mol/mol in patient samples and from 0.3 to 1.0 mol/mol in controls of healthy donors. The bilirubin content of patient samples was 0.09–3.56 mg/g albumin. HSA of healthy volunteers contained 0.00–0.04 mg bilirubin per g albumin. Upon binding of dansylsarcosine (DS) $K_d$ ranged from 13–28 µmol/L in patients' HSA and from 12–18 µmol/L in HSA of healthy donors. The distribution of reduced and oxidized fractions of patients' HSA was 36–58% HMA, 31–51% HNA1, and 8–14% HNA2. Healthy controls' HSA was distributed in 63–72% HMA, 18–29% HNA1, and 6–10% HNA2, respectively. Demographic data of patients and detailed characterization of prepared patient HSA samples are shown in Table 1. Plasma variables related to liver disease of all patients are shown in Supplementary Table 1.

We observed a correlation of HSA characteristics such as FA and bilirubin content, redox state, and the severity of liver disease

**Table 1 Demographic data of patients with chronic liver failure and healthy control persons and characterization of thereof prepared HSA samples.**

| Pat. | Sex | Age | MELD | FA/HSA | Bilirubin/HSA | $K_d$ | HSA redox state | | |
|---|---|---|---|---|---|---|---|---|---|
| | | Years | | mol mol$^{-1}$ | mg g$^{-1}$ | µmol L$^{-1}$ | HMA % | HNA1 % | HNA2 % |
| P1 | Male | 51 | 29.1 | 1.0 | 0.81 | 28 | 45 | 42 | 14 |
| P2 | Female | 69 | 26.1 | 3.0 | 0.09 | 15 | 36 | 51 | 13 |
| P3 | Male | 81 | 11.2 | 1.5 | 0.09 | 25 | 52 | 37 | 11 |
| P4 | Male | 56 | 13.0 | 1.4 | 0.18 | 22 | 55 | 34 | 11 |
| P5 | Female | 42 | n.a. | 2.1 | 0.18 | 28 | 58 | 34 | 8 |
| P6 | Male | 58 | 12.8 | 0.6 | 0.41 | 17 | 52 | 36 | 12 |
| P7 | Male | 48 | 24.9 | 0.6 | 3.56 | 20 | 57 | 31 | 12 |
| P9 | Male | 69 | 7.4 | 1.2 | 0.13 | 13 | 48 | 41 | 11 |
| C1 | Female | 35 | n.a. | 0.6 | 0.02 | 18 | 72 | 23 | 6 |
| C2 | Male | 55 | n.a. | 0.3 | 0.02 | 12 | 69 | 25 | 6 |
| C3 | Male | 28 | n.a. | n.a. | n.a. | n.a. | 63 | 29 | 8 |
| C4 | Female | 23 | n.a. | 1.0 | 0.04 | n.a. | 72 | 18 | 10 |

*MELD* model for end-stage liver disease, *FA* fatty acids, *n.a.* not available.

with the conformation of albumin. We observed only small differences in the conformation of in vitro reduced and oxidized albumin fractions.

**Redox state of Cys-34 does not alter the HSA conformation**. To determine conformational variability upon HSA Cys-34 oxidation, which may impair its transport function[27,28], we prepared three fractions of HSA according to the Cys-34 redox state from a healthy donor (see "Methods" section). The three fractions of HSA are (i) HMA, the reduced form with a free thiol group on Cys-34, (ii) HNA1 oxidized, with a disulfide bond to cysteine,

homocysteine, or glutathione, and (iii) HNA2 with Cys-34 oxidized to sulfinic or sulfonic acid[17,18]. We determined conformational variability between HSA and HSA fractions by using size exclusion chromatography (SEC) coupled SAXS with multi-angle light scattering (SEC-SAXS-MALS) (Fig. 1a). SEC prior to SAXS separated small amounts of self-associating HSA dimers, which had not been considered in the further analysis (Supplementary Fig. 1a). Analyzing the SEC-SAXS chromatograms from the monomeric peaks demonstrated a symmetrical peak for all HSA fractions and control samples (healthy donor HSA). Calculated MW using MALS and $R_g$ values for each SAXS frame across the peak further indicate sample homogeneity (Fig. 1a).

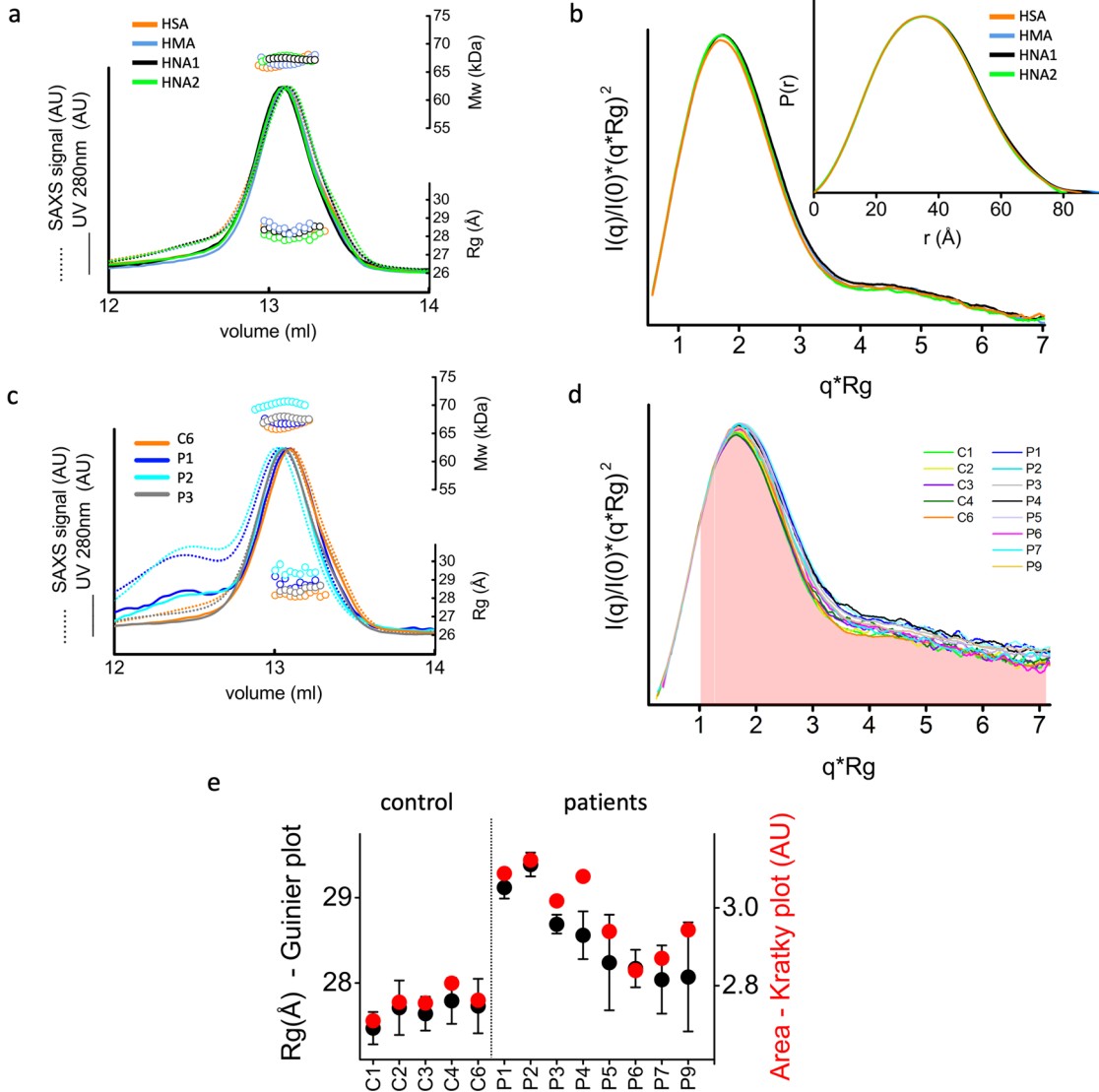

**Fig. 1 Experimental SAXS curves suggest conformational differences between healthy and patients' HSAs. a** SEC-SAXS-MALS chromatograms for HMA (blue), HNA1 (black), HNA2 (green), and control HSA sample (orange). Solid lines represent the UV signal. Dot lines represent integrated SAXS signals in arbitrary units (AU). In contrast, symbols represent molecular mass determined by MALS (top) and $R_g$ values determined by Guinier plots (bottom) for each collected SAXS frame versus elution volume. **b** Experimental SAXS curves plotted as normalized Kratky plots and $P(r)$ functions from the experimental SAXS curves of HMA, HNA1, HNA2, and control HSA sample. For better visualization, the Kratky plot was smoothed by the Savitzky-Golay filter. Corresponding unfiltered SAXS curves are shown in Supplementary Fig. 2a. **c** SEC-SAXS-MALS chromatograms for three patient samples (P1-blue, P2-cyan, P3-gray) in comparison to the control HSA (C6-orange) sample. Solid lines represent the UV signal. Dot lines represent integrated SAXS signals in arbitrary units (AU). In contrast, symbols represent molecular mass determined by MALS (top) and $R_g$ values (bottom) for each collected SAXS frame versus elution volume. **d** Experimental SAXS curves of control and patient samples plotted as a normalized Kratky plot. For better visualization, the Kratky plot was smoothed by the Savitzky-Golay filter. Corresponding unfiltered SAXS curves are shown in Supplementary Fig. 2b. The range of the Kratky plot used to calculate the area under the curves is highlighted with a red surface. **e** $R_g$ values together with SD (black dots) for control, and patient samples, determined from the Guinier plot shown in Supplementary Fig. 2c. Areas under normalized Kratky plots shown in panel **d** (red dots).

SAXS profiles for the monodisperse monomeric state of the HMA, HNA1, HNA2, and control HSA (Fig. 1b and Supplementary Fig. 2a) were transformed into the pair distribution functions ($P(r)$) that were further normalized at the maxima (Fig. 1b inset). Similar shape and broadening of the $P(r)$ functions indicate similar overall conformation of HSA fractions in solution. However, a small peak shift in the Kratky plots (Fig. 1b) and a minor increase in the $R_g$ values of HMA and HNA1 (Supplementary Table 2) suggest an altered local fold of the reduced and oxidized HSA.

The fact that the redox state has only minimal impact on the solution state of HSA leads us to further visualization of HSA conformations related to liver diseases.

**HSA biochemistry correlates to its solution state**. We determined conformational variability between "healthy" and "patient" HSAs using an identical approach as shown above for HSA fractions (Fig. 1c and d) and further correlated with the content of FA, bilirubin, and redox state (Table 2).

SAXS curves for the monodisperse monomeric state of HSA samples (Supplementary Fig. 2b) were transformed into the normalized Kratky plots (Fig. 1d). The shift in the peak of normalized Kratky plot towards larger $q \times R_g$ values indicates a conformational change of patient HSAs. By calculating areas under normalized Kratky plots we quantitatively compared these changes. The correlation of these areas with the $R_g$ values determined by the Guinier plot for SAXS curves (Supplementary Fig. 2c) shows increased dynamicity of patient HSAs (Fig. 1e, Supplementary Table 2).

To ensure that imperfect buffer subtraction or different signal-to-noise ratios do not impact SAXS curves comparison we also compared $P(r)$ functions. A closer comparison of $P(r)$ functions shows broader $P(r)$ for the patient samples relatively to more compact $P(r)$ functions of a healthy donor and commercial HSA (Fig. 2a). The broadening of patient $P(r)$ functions indicates open HSA conformation and can be correlated with the FA interaction, as shown in the structure of HSA complexed with myristic acid[8] (PDBID 1BJ5, Fig. 3a).

Next, we correlated the broadening of $P(r)$ with the relation to the FA, bilirubin content, and redox state. $P(r)$ functions of HSA from all patients and controls are shown in Fig. 2a. The $P(r)$ broadening was determined by the $r$ distances at $P(r)$ intensity = 0.2. These $r$ values were correlated to the characterized, ranked variables of HSA. $P(r)$ broadening correlates with the FA content of HSA. Furthermore, maximal dimensions ($D_{max}$) determined from the $P(r)$ functions (Supplementary Table 2) correlate with FA and bilirubin content of HSA, as well as MELD score (Table 2). We

found correlations with HNA1 and HNA2 and an inverse correlation with HMA content for $P(r)$ broadening and $D_{max}$ values (Table 2).

Next, we applied the volatility of ratio ($Vr$) difference metric, which provides a quantitative and superposition-independent evaluation of SAXS curves or $P(r)$ functions[29]. By plotting a diagonally symmetric heat map in which each matrix element (cell) quantifies the pairwise agreement between the SAXS datasets, color mapped from red (similar) to white (different) (Fig. 2b, c). $Vr$ values displayed in a heat map derive from the ratio between two SAXS curves or two $P(r)$ functions. Advantages of $Vr$ over Chi2 metric can be subtle[29], however, $Vr$ highlights key distinctions that are missed by the Chi2 metric (Fig. 2b).

The heat maps for SAXS curves (Fig. 2b) show similarities between all patients, but also indicate some level of dissimilarities in between control samples. We believe that these dissimilarities are related to the different signal-to-noise ratios in the SAXS curves. Conversely, the heat map for $P(r)$ functions (Fig. 2c) shows a close similarity between control HSAs. The control HSA offers distinct dissimilarities to all $P(r)$ functions derived for patients' HSA, except for the patient sample P6. However, the HSA redox state does not alter the overall HSA conformation in a healthy donor (Fig. 1a, b). Furthermore, this suggests that the pathological content of FA/bilirubin in patients may be related to the HSA redox state. No correlations were found with $K_d$ of DS binding to HSA. In contrast, we found a correlation of the Kratky plot area with age. This may be due to the strong correlation with HNA1 which is increasing with age[15].

**Patient HSA shows a distinct opening of domain I**. The broadening of patients' $P(r)$ functions suggests the transition of HSA to a more open conformation. Tilt and twist of N-terminal and C-terminal HSA regions (domain I and III) of lipid-bound HSA structure[8] in relation to the compact lipid-free HSA structure[30] (Fig. 3a) reflects the type of conformational rearrangement that HSA may undergo in the patient samples. To validate whether the HSA undergoes this type of allostery, we used SAXS profiles to interrogate conformational changes of commercial HSA (C6), a healthy donor (C1), and two patients HSA (P1 and P2). The identical SAXS profiles of commercial and control HSA confirm a very similar solution state. The scattering differences in the small-angle range (q 0.1–0.2 Å$^{-1}$) between commercial and patients' HSA indicate changes in the overall rearrangement of patient HSA domains (Fig. 3b) rather than changes in the local fold. To visualize HSA allostery, the conformational sampling protocol was applied using normal mode analysis (NMA)[31].

**Table 2 Spearman's rank correlation coefficients (r) and associated P values of age (control persons + patients), MELD score (patients), and HSA variables (control persons + patients) with ranked distances determined from P(r) function at intensity P (r) = 0.2 (see Fig. 2a), maximal dimension derived from P(r) function (Dmax) (see (Supplementary Table 2) and area under the Kratky plot (see Fig. 1d and Supplementary Table 2).**

|  |  | Age | MELD | Ligands | | DS binding | Redox state | | |
|---|---|---|---|---|---|---|---|---|---|
|  |  |  |  | FA | Bilirubin | $K_d$ | HMA | HNA1 | HNA2 |
| $P(r)$ broadening | $r$ | 0.372 | 0.487 | 0.774 | 0.524 | 0.249 | −0.658 | 0.666 | 0.609 |
| ($P(r) = 0.2$) | $P$ | 0.232 | 0.271 | **0.007** | 0.101 | 0.484 | **0.023** | **0.021** | **0.039** |
| Maximal dimension ($D_{max}$) | $r$ | 0.385 | 0.821 | 0.600 | 0.800 | 0.273 | −0.734 | 0.713 | 0.769 |
|  | $P$ | 0.215 | **0.034** | 0.056 | **0.005** | 0.448 | **0.009** | **0.012** | **0.005** |
| Kratky plot area | $r$ | 0.644 | 0.571 | 0.800 | 0.645 | 0.358 | −0.832 | 0.846 | 0.727 |
|  | $P$ | **0.027** | 0.200 | **0.005** | **0.037** | 0.313 | **0.001** | **0.001** | **0.010** |
|  | $n$ | 12 | 7 | 11 | 11 | 10 | 12 | 12 | 12 |

$P < 0.05$ was considered statistically significant; $n$, exact sample size. Significant correlations are highlighted in bold.

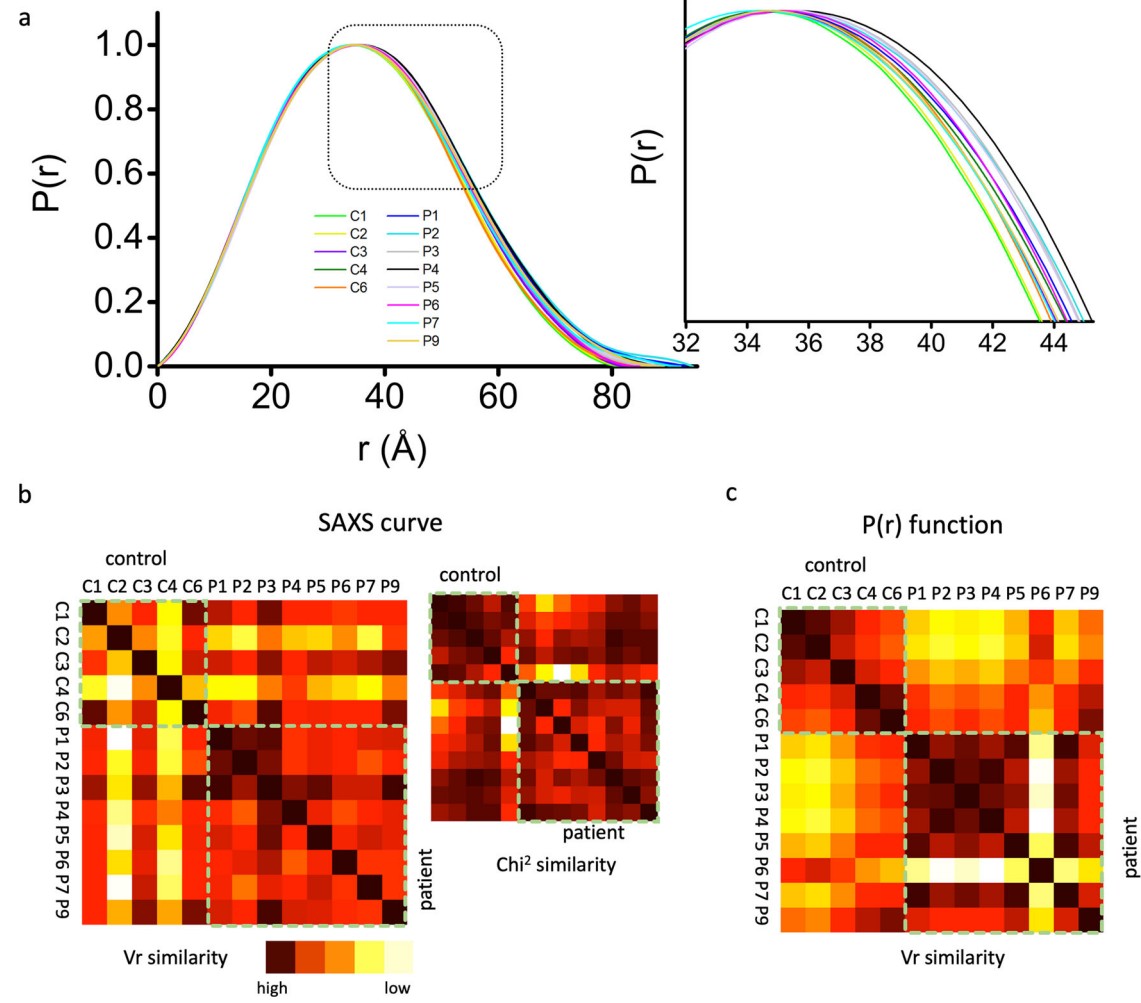

**Fig. 2 Broadening of *P(r)* functions correlate between patient and control samples. a** *P(r)* functions and zoom-in section show the shift in *P(r)* maxima and distinct broadening for patient samples. **b** A measure of structural similarity between SAXS curves scored by the volatility of ratio (*Vr*) and Chi$^2$ shown in the inset. **c** A measure of structural similarity between *P(r)* functions scored by the volatility of ratio (*Vr*). Scores were assigned a gradient color with a red-high agreement and white-low agreement.

Lipid-free HSA structure (PDBID: 1AO6[30]) was divided into three domains representing domain I, II, and III (1–175, 176–495, and 495–585), and constrained NMA sampling was applied to optimize the position of each region relative to the each other including normal mode movement inside each region. The conformational variability between the top-scoring models points out the uncertainty of SAXS based modeling (Fig. 3c). Thus, these atomistic models cannot be understood as unique static structures. Conformational variability of the domain I and III in the top-scoring models of the patient samples is larger than that of control samples (Fig. 3c) and further validate increased dynamicity in patient HSA. We visualized quantitative differences between control and patient conformers by superimposing crystal structure with the best fit model. The best SAXS fit model was found by searching the one-state model[32] from the pool of eight top-scoring models derived from NMA conformational sampling. The goodness of fit ($\chi^2$) between theoretical SAXS curves from the atomistic model and experimental data[33] is merit for the selection. The selected models for the commercial and 'healthy' HSA show only a small improvement in the goodness of fit relative to the crystal structure ($\chi^2$ 3.3 vs. 1.5–commercial HSA; $\chi^2$ 2.2 vs. 1.4 healthy donors, Fig. 3c). The models of commercial and 'healthy' HSA show only a small tilt in the domain I and III relative to the crystal structure (Fig. 3d), indicating the inherent

dynamicity of these regions in the absence of stabilizer in solution.

On the other hand, the best fit model of patients' HSA shows the distinct movement of both domains that resemble the allosteric transition observed in the lipid-bound HSA structure (Fig. 3c). The domain III movements can be as large as 16 Å shown for the patient P2 HSA. This analysis indicates that patient HSA adopts an open conformation that correlates with the FA lipid state and offers the capability of SAXS to determine conformational changes upon FA binding.

## Discussion

The promise of macromolecular structural biology is to visualize structures that unveil critical functional mechanisms. This promise is often not fully realized due to missing or incomplete knowledge of functional conformations assessed outside the physiological condition. On the other hand, SAXS, which can be high-throughput[34] and directly measures thermodynamic solution-state conformational changes and assemblies, is resolution limited. Yet, SAXS can be combined with an X-ray crystal structure to interrogate atomic level information[35]. Here we assess practical and robust methods to join SAXS with existing X-ray crystal structures as applied to elucidating the conformational

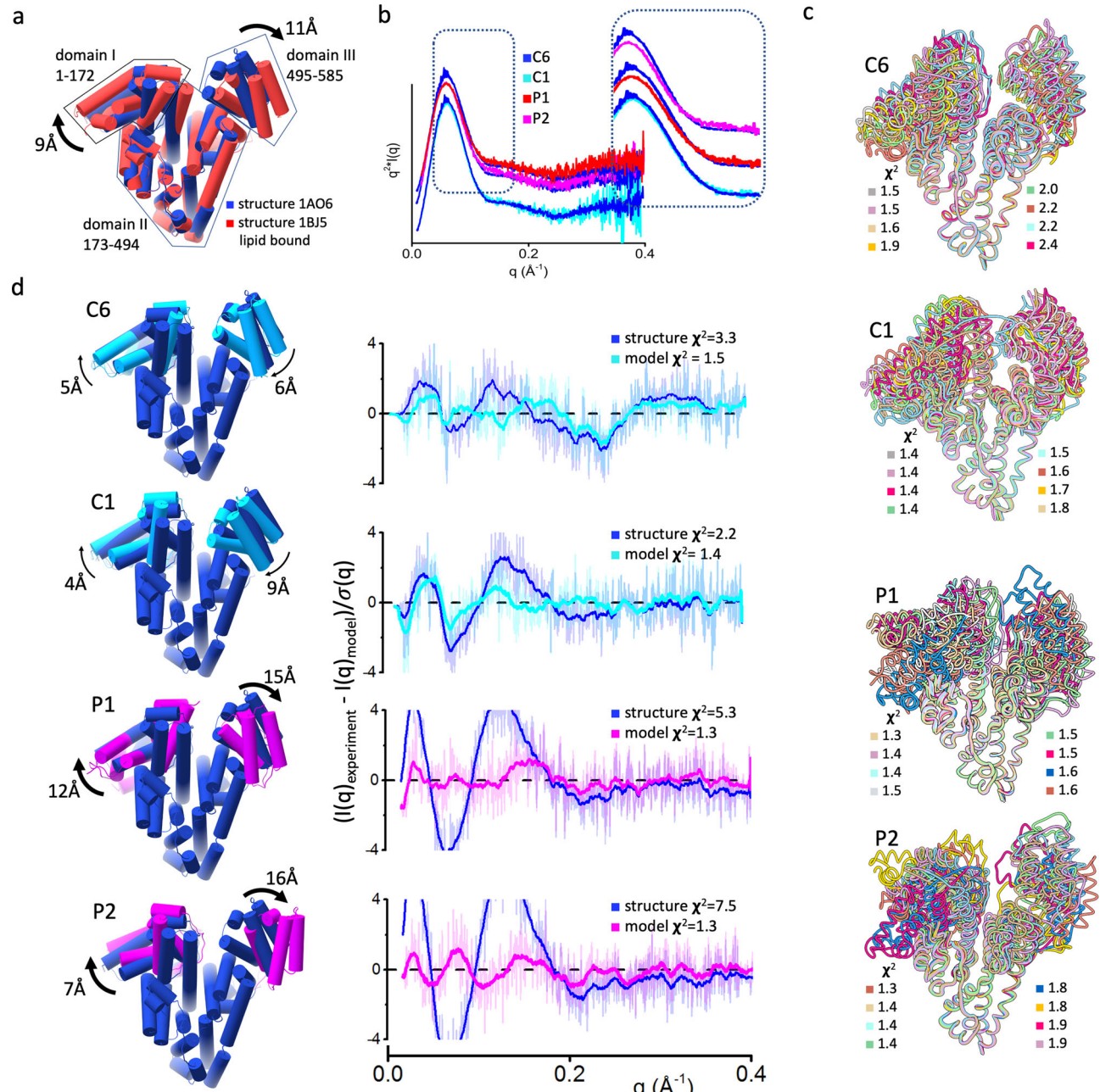

**Fig. 3 Patients' HSA undergoes a distinct movement of domains I and III. a** Superposition of lipid-free (blue) and lipid-bound (red) HSA (PDBID 1AO6 and 1BJ5) crystal structure shows the opening of domains I and III upon lipid binding. **b** Comparison of Kratky plot for commercial HSA (C6–blue) with a healthy donor (C1–cyan) and patients' HSA (P1 and P2–red and magenta) suggest overall structural changes. **c** Eight top-scoring atomistic models for commercial HSA (C6), healthy donor (C1), and patients' HSA (P1 and P2). Atomistic models were superimposed on domain II. The goodness of SAXS fit ($\chi^2$) for each model is shown in the figure legend with the corresponding color code. **d** The best fit atomistic models of commercial HSA (C6–cyan), a healthy donor (C1–cyan), and two patients (P1, P2–magenta) superimposed on to the lipid-free HSA structure (PDBID 1AO6–blue). The left panel shows residuals and goodness of fit values ($\chi^2$) between experimental and theoretical SAXS calculated from the atomistic models or lipid-free HSA structure (PDBID 1AO6–blue).

variability of HSA between healthy donors and patients with liver disease.

Chronic liver disease is often associated with reduced HSA synthesis, increased oxidative stress[16], altered lipid and lipoprotein patterns[36,37], and elevated bilirubin levels that consequently increase the amount of oxidized HSA fractions, as well as the molar ratio of FA and bilirubin per albumin. Bilirubin is a variable included in the model of end-stage liver disease (MELD) score reflecting the degree of severity of the chronic liver disease.

Our results show increased levels of oxidized albumin, as well as of FA and bilirubin associated with patients' HSA accompanied by a distinct structural opening of HSA domains I and III. Now there is the question of whether the altered HSA conformation is functionally impaired. Altered HSA conformation may decrease affinity to the GP60 (Albondin) receptor that enables transcytosis of albumin, but conversely increases the degradation of altered albumin through GP18/GP30[38,39]. Furthermore, SPARC (secreted protein acidic and rich in cysteine) facilitates transcytosis of

FA-loaded albumin through endothelial cells[40]. HSA saturated with long-chain FA has a reduced affinity for the neonatal Fc receptor (FcRn) that protects albumin from degradation and promotes its recycling[41]. This suggests that structural changes of HSA induced by FA-binding may affect HSA recycling, degradation, and receptor-mediated uptake and, consequently, influence the mean lifetime of HSA[42,43].

Moreover, the changes in HSA conformation by pathologically high levels of FA and bilirubin, as seen in chronic liver disease, could cause faster irreversible post-translational modifications such as oxidation, glycation, truncation, and dimerization and thereby reduce the so-called "effective albumin concentration"[44]. This may have an additional negative impact on patients with already reduced albumin plasma concentrations. Somewhat surprisingly, we observe only very small conformational differences in in vitro prepared, reduced, and oxidized HSA fractions HMA, HNA1, and HNA2.

However, the degree of oxidation correlates with the maximal dimension of HSA ($D_{max}$) and its dynamicity measured by the Kratky plot and $P(r)$ broadening. This might implicate that not the oxidation of HSA but the simultaneous enhancement of ligand loading is critical in chronic liver disease.

Interestingly, oxidized albumin constitutes a prognostic marker for the survival of end-stage liver disease patients[11] and is functionally impaired[16]. Maybe this effect is induced by molecular rather than overall conformational changes of oxidized HSA structure. We demonstrate, to the best of our knowledge, for the first time that conformation and flexibility of patient HSA are changed compared to healthy donor HSA. The latter has inherent flexibility with the dynamic movement of domains I and III. Our experimental results agree with previous simulation studies[45,46] and further confirm that the inherent flexibility of HSA has functional importance. This HSA plasticity can be inhibited by stabilizers, like those used in clinical HSA infusion solutions and endogenous ligands, and often leads to impaired transport function and drug binding capacity[47].

These collective results support a structural understanding of HSA dynamicity and suggest that increased FA and bilirubin levels impair its functional role. Specifically, integrating SAXS measurements with atomistic modeling provides data-based dynamic models that indicate how the flexibility of HSA domains I and III control its function.

## Methods

### Ethics approval and consent to participate
Sample collection was conducted at the Department of Internal Medicine, Division of Gastroenterology and Hepatology, Medical University of Graz, between January 2017 and December 2018. It was approved by the local ethics committee (registered at the Office for Human Research Protections of the US Departments of Health and Human Services: IRB00002556) (patients: 29-040 ex 16/17, controls: 29-460 ex 16/17) and conducted following the Declarations of Helsinki. All patients and healthy control persons gave written informed consent before their inclusion in the study.

### Plasma samples
For plasma preparation, ethylene diamine tetra acetic acid (EDTA) blood was drawn from each donor. Blood samples were centrifuged immediately at 4 °C at the in-house laboratory of the Division of Gastroenterology and Hepatology to separate plasma. Plasma was aliquoted and stored at −80 °C until further analyses.

Albumin of patients with chronic liver failure was evaluated ($n = 8$), and the results were compared to a control group ($n = 4$). For both groups, the inclusion criteria comprised age above 18 years and ability to provide informed consent. For patients with chronic liver failure, exclusion criteria were malignant ascites, presence of hepatocellular carcinoma or advanced extrahepatic neoplasia, nephrotic syndrome, pregnancy or lactation, and albumin infusion (>80 g) within the last 48 h.

We included 8 patients with decompensated cirrhosis admitted to our GI ward for treatment of tense ascites refractory to diuretics. All patients underwent large-volume paracentesis followed by albumin infusion (10 g per liter ascites). Blood samples for albumin preparation were collected prior to paracentesis and subsequent albumin infusion, respectively.

### Commercial albumin
Commercial albumin from human serum was purchased from Sigma Aldrich (Sigma HSA) and served as additional control C6.

### Sample preparation
HSA from patients with chronic liver failure and healthy donors were prepared by loading EDTA-plasma to a HiTrapTM Blue HP column (GE Healthcare, Solingen, Germany). After equilibrating the column with 50 mM potassium dihydrogenphosphate (pH 7.0), HSA was eluted with 50 mM potassium dihydrogenphosphate solution containing 1.5 M potassium chloride, and the respective fractions were collected. Pooled fractions were dialyzed against distilled water overnight at 4 °C, applied to a PD-10 desalting column (GE Healthcare), and eluted with phosphate-buffered saline (PBS; 137 mM NaCl, 2.7 mM KCl, 10 mM Na$_2$HPO$_4$, 1.8 mM KH$_2$PO$_4$, pH 7.4). Purified HSA was stored at −80 °C until further analyses. Sigma HSA was dissolved in PBS and also stored at −80 °C.

We prepared reduced and oxidized albumin fractions (HMA, HNA1, HNA2) from EDTA plasma of one female healthy donor aged 35 years. In brief, whole blood was drawn and incubated with 1 mg/mL N-acetylcysteine (Sigma Aldrich) for 1 h at room temperature (RT). This leads to a reduction of HNA1, resulting in mainly HMA. HMA enriched plasma was obtained upon centrifugation, and HMA was prepared as described above for HSA. To obtain HNA1, the purified HMA was incubated with 17 mmol/L cysteine at 37 °C for at least 24 h. Similarly, HNA2 was prepared by incubating HMA with 45 mmol/L hydrogen peroxide (H$_2$O$_2$) for 1 h at RT. Complete oxidation was examined by HPLC and residual cystine or H$_2$O$_2$ was removed by dialysis against distilled water. Each fraction was applied to a PD-10 desalting column and eluted with PBS. Purified HMA, HNA1, and HNA2 fractions were stored at −80 °C until further analyses.

### Albumin redox state
The distribution of HMA, HNA1, and HNA2 in prepared HSA samples of patients and controls was determined by high-performance liquid chromatography (HPLC) described by Hayashi et al.[48] In brief, plasma samples were diluted 1:100 with 0.1 M sodium phosphate, 0.3 M sodium chloride, pH 6.87, filtered through a Whatman 0.45 µm nylon filter (Bartelt Labor and Datentechnik, Graz, Austria), and 20 µL of the respective diluted sample were injected into the HPLC system, using a Shodex Asahipak ES-502N 7 C anion exchange column with 50 mM sodium acetate, 400 mM sodium sulfate, pH 4.85, as mobile phase. A gradient of 0 to 6% ethanol and a 1 mL/min flow rate was used for elution. The column was kept at 35 °C and detection was carried out by fluorescence at 280/340 nm. Quantification was based on the individual peaks' heights, as determined by EZ Chrome Elite chromatography software (VWR, Vienna, Austria).

### Fatty acids (FA) content
Non-esterified FA content in HSA preparations was determined by an in vitro enzymatic colorimetric method (FUJIFILM Wako Diagnostics, Neuss, Germany) according to the manufacturer's instructions. Briefly, 10 µL of each sample were mixed with 150 µL reagent 1 and incubated for 10 min at 37 °C. Subsequently, 75 µL reagent 2 was added and incubated for an additional 10 min at 37 °C. Extinction was measured at 546 nm, and FA content was normalized to albumin concentration, determined by Albumin (BCG) Assay Kit (Abcam), in the respective sample.

### Bilirubin content
HSA bound bilirubin was measured by an in vitro colorimetric test assay (Human Biochemica und Diagnostica GmbH, Wiesbaden, Germany). In brief, 30 µL of the respective HSA sample was mixed with 200 µL reagent and incubated for 10 min at RT. Extinction was measured at 546 nm and bilirubin content was normalized to albumin concentration in the respective sample.

### Dansylsarcosine (DS) binding
Prepared HSA was diluted with phosphate-buffered saline (PBS) to a final concentration of 10 µmol/L, and 100 µL of this dilution was applied to a flat-bottomed, half area, polystyrene, black 96-well plate (Greiner Bio-One, Kremsmuenster, Austria). Successively, 100 µL of different concentrations of DS (0.625 µmol/L, 1.25 µmol/L, 2.5 µmol/L, 5 µmol/L, 10 µmol/L, 20 µmol/L, 40 µmol/L, 80 µmol/L, 160 µmol/L, 320 µmol/L, and 640 µmol/L) were added and the plate was shaken gently before incubation for 20 min at 37 °C. The resulting fluorescence was measured using FLUOstar OPTIMA microplate reader (BMG LABTECH, Ortenberg, Germany). Fluorescence excitation and emission detection were performed at 355 nm and 460 nm, respectively. Binding curves were fitted applying non-linear regression using GraphPad Prism Software (Version 5; GraphPad Software, San Diego, CA, USA) after subtraction of non-specific binding. Dissociation constants ($K_d$) were calculated at DS concentrations, where half-maximal fluorescence was reached.

### Calculation of model for end-stage liver disease (MELD)-score
The severity of chronic liver failure was estimated by calculating the original MELD score as follows: MELD = $10 \times (0.957 \times \ln(\text{creatinin}) + 0.378 \times \ln(\text{bilirubin}) + 1.12 \times \ln(\text{INR}) + 0.643)$.

### Small-angle X-ray scattering and multi-angle light scattering data acquisition in line with size-exclusion chromatography (SEC-SAXS-MALS)
For SEC-SAXS-MALS experiments, 60 µL of samples containing 5–10 mg/mL of HSA were prepared in 20 mM Tris pH 7.4, 150 mM KCl, 2% glycerol. HSA samples were further filtered through 300 kDa cut-off ultra-filters (Amicon) that eliminate most of the HSA

higher oligomers and further dilute the sample. SEC-SAXS-MALS were collected at the ALS beamline 12.3.1[49]. X-ray wavelength was set at $\lambda = 1.127$ Å and the sample to detector distance was 2100 mm, resulting in scattering vectors, $q$, ranging from 0.01 Å$^{-1}$ to 0.4 Å$^{-1}$. The scattering vector is defined as $q = 4\pi\sin\theta/\lambda$, where $2\theta$ is the scattering angle. All experiments were performed at 20 °C and data were processed as described[34]. Briefly, a SAXS flow cell was directly coupled with an online Agilent 1260 Infinity HPLC system using a Shodex KW 803 column. The column was equilibrated with running buffer (20 mM Tris pH 7.4, 150 mM KCl, 2% glycerol) with a 0.5 mL/min flow rate. Fifty-five microliters of each sample was run through the SEC and three second X-ray exposures were collected continuously during a 30 min elution. The SAXS frames recorded prior to the protein elution peak were used to subtract all other frames. The subtracted frames were investigated by the radius of gyration ($R_g$) derived by the Guinier approximation $I(q) = I(0)\exp(-q^2 \times R_g \times 2/3)$ with the limits $q \times R_g < 1.5$. The elution peak was mapped by comparing integral ratios to background and $R_g$ relative to the recorded frame using the program SCÅTTER. Uniform $R_g$ values across an elution peak represent a homogeneous monomeric state of HSA. Final merged SAXS profiles (Supplementary Fig. 1a and b), derived by integrating multiple frames across the elution peak, were used for further analysis, including a Guinier plot which determined the aggregation free state (Supplementary Fig. 1c). In determining the linear portion of the Guinier region required to obtain an accurate Rg value, the first few points of the curve often require elimination due to limitations in data collection. This leads to a conservative truncation of the low $q$ range to ~0.01 Å$^{-1}$ before continuing analysis. Kratky plot was normalized using $R_g$ and Intensity values at the zero angles [I(0)] determined from the Guinier plot. The area under the normalized Kratky plot was calculated in the $q \times R_g$ range between 1 and 7 (see Fig. 1d and Supplementary Table 2). The program GNOM[50] was used to compute the pair distribution function ($P(r)$) (Fig. 1b and 2a). The distance $r$ where $P(r)$ approaches zero intensity identifies the macromolecule's maximal dimension ($D_{max}$, Supplementary Table 2). $P(r)$ functions were normalized at the maxima.

The SAXS flow-cell was also connected inline to a 1290 series UV-vis diode array detector measuring at 280 and 260 nm, 18-angle DAWN HELEOS II multi-angle light scattering (MALS), and quasi-elastic light scattering (Wyatt Technology), and Optilab rEX refractometer (Wyatt Technology). System normalization and calibration were performed with bovine serum albumin using a 45 μL sample at 10 mg/mL in the same SEC running buffer and a dn/dc value of 0.175. The light scattering experiments were used to perform analytical scale chromatographic separations for MW and hydrodynamic radius (Rh) determination. UV, MALS, and differential refractive index data were analyzed using Wyatt Astra 7 software to monitor sample homogeneity across the elution peak complementary to the above-mentioned SEC-SAXS signal validation.

**Solution structure modeling**. The lipid-free HSA structure[30] (PDB 1AO6) was used to build an initial atomistic model, missing four N-terminal and three C-terminal residues were added using MODELLER[51]. A normal mode analysis (NMA) approach SREFLEX[31] was used to determine the movement of domain I (residues 1–172) and domain III (residues 495–585) in solution with the central domain II of the molecule fixed. SREFLEX systematically explored conformational space for both domains I and III simultaneously by optimizing goodness of fit to the experimental scattering curves for control C1, C6, and patients P1, P2. Eight top-scoring SREFLEX-generated models were then pooled together and the final search for the best fit model was done by FOXS[33,52].

**Statistics and reproducibility**. Variables are presented as single values, ranges, or means ± standard deviations. Two-tailed Spearman's rank correlation analysis was performed using GraphPad Prism software (Version 9) to explore the relations between variables of 7 to 12 individual samples (patients and healthy controls). $P < 0.05$ was considered to indicate statistical significance.

**Reporting summary**. Further information on research design is available in the Nature Research Reporting Summary linked to this article.

## Data availability

SEC-SAXS-MALS data, including atomistic models, are deposited in the Simple SAXS data bank (https://simplescattering.com/) and SASBDB data bank (https://www.sasbdb.org/). IDs are listed in Supplementary Table 2. Any other information can be obtained from the corresponding author upon reasonable request.

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

## Acknowledgements

SAXS data collection at SIBYLS Beamline of the Advanced Light Source, a U.S. DOE Office of Science User Facility under Contract No. DE-AC02-05CH11231, is funded through DOE BER Integrated Diffraction Analysis Technologies (IDAT) program and NIGMS grant P30 GM124169-01, ALS-ENABLE. M.H is further supported by the NIH grants P01 CA092584. ALBUS Award 2016 to R.E.S. The excellent technical assistance of Martina Mairold and Doris Payerl is highly appreciated.

## Author contributions

M.H., K.O., and A.K. conceived the project. R.E.S. was responsible for the recruitment of patients and plasma collection. M.H., D.J.R., and M.P. planned and performed experiments. M.H. created Figures. M.H., D.J.R., K.O., A.K., and M.P. analyzed the data. M.H., K.O., M.P., and V.H.F. wrote the manuscript. All authors contributed to the discussion and interpretation of the data.

## Competing interests

The authors declare no competing interests.
