## [Transparent Peer Review File · Communications Biology]

Reviewers' comments:

Reviewer #1 (Remarks to the Author):

Paar M. and all propose in this paper a novel approach based on Small Angles X-Rays Scattering to study conformations of HSA coming from patients affected by liver disease. This article constitutes an original approach to correlate clinical observations and a structural data. HSA has been widely studied with analytical methods such as chromatography and mass spectrometry (i.e. Naldi M. Journal of Pharmaceutical and Biomedical Analysis 144(2017) 138-153) to characterize the most abundant modifications affecting HSA structure. The originality of this paper resides in the use of structural method in solution such as SAXS to sense the protein conformation in solution.

In summary, the article is very well written, easy to read and which complements the bibliography on the subject well. However, although the authors had a rigorous approach with the SAXS by using an HPLC online and molecular modeling with normal modes, I have some questions and remarks about the SAXS described in my reports.

- the first question concerns the data acquisition. Why the data are truncated at 0.01 \AA^{-1} ? The data were recorded on a synchrotron beamline and with the setup described in the paper we can expect to obtain a q_{\min} value close to 0.004 \AA^{-1} at 2.1m with $\lambda = 1.127 \text{ \AA}$ (i.e. with a laboratory source at $\lambda = 1.54 \text{ \AA}$ and $d = 1.2\text{m}$ the $q_{\min} = 0.005$ and $q_{\max} = 0.5 \text{ \AA}^{-1}$). In the paper the most important differences are located in small angles regions which give the difference at high distances in the $P(r)$ representation. Although the q range used by the authors are theoretically sufficient to study a small protein such as HSA, I am surprising to see data cut at 0.01 \AA^{-1} . The Guinier's representations shown in sup. data are good but what happens if you extend the plot to the smallest q values? Did you have some problems with radiation damages or concentration effects (due to the low concentration of salt) which could affect the beginning of the curves? The variations observed in the $P(r)$ plot are minimal and the beginning of the SAXS curves strongly influences the maximal distance of $P(r)$ function.

- the second point is a remark concerning the figure 1 where are plotted chromatogram. Instead of to show the integrated SAXS signal which is effectively proportional to the concentration, wouldn't it be more relevant to show the concentration of the protein $[C]$ and in a same time, the value of $I(0)$ divided by the concentration $[C]$? As much as $I(0)$ depends of the concentration and the size of the object, it is a good indicator to check the homogeneity of the elution peak.

-the third point is also a remark and concerns the use of the commercial HSA provided by Sigma. At our laboratory, we use commonly the same HSA provided by this company and we observe always the dimer peak during elution profile in the PBS buffer conditions with a similar column (Bio-Sec3-300 Agilent). You observe only the dimer with the patients P1 and P2 ?

-the fourth point concerns the molecular modelling. The approach combining the comparison with crystallographic structures and all-atom models generated by normal modes is relevant. SAXS is a powerful tool to discriminate structures each other but I'm always reserved when it comes to doing the opposite operation. The approach with normal modes allows to generate possible conformations with a non-aberrant global energy as we often find in modelling algorithms against SAXS data. However, the SAXS stays a low resolution structural method with an important uncertainty when models are generated by molecular modeling algorithm. I do not question the results found by the authors, I am convinced by their results thanks to the rigor of their work. But it will be possible to evaluate this uncertainty by superimpose 10 (or more) structures just shown in ribbon (just backbone without secondary structures) generated with normal nodes with a chi value close to each other and by highlighting the best model? In this way I suppose we can judge the degree of imprecision of the modeling, the aim being to show the reader that even if the SAXS is a low resolution method it can give relevant indications on structural modifications but with a relative precision that can be easily

evaluated. Maybe this could be the subject of an additional figure in the material and method, not in the main text.

In conclusion, the authors provide a very good and original work combining clinical profiles of patient affected by liver diseases, biochemical measurements, structural information coming from X-ray measurement in solution and molecular modeling. The conclusions are coherent and give a new insight in the study of liver's pathology most specifically on the knowledges of HSA properties.

Reviewer #2 (Remarks to the Author):

The manuscript describes the difference in the conformational state of HSA in plasma of healthy people and patients with pathological liver diseases. The correlation between HSA structural changes with some pathological conditions, such as redox state of HSA, redundancy of fatty acids and bilirubin in human plasma, is reported. The authors employ SAXS on real clinical samples, therefore this work not only provides insights into the conformational transitions of the most abundant protein in human blood plasma but also has concrete practical implications. Further, the authors show that the HSA quaternary structure from patients with chronic liver diseases can be used as prognostic markers. To describe the conformational change, the authors utilize the formal SAXS analysis and normal mode analysis in program SREFLEX and find the increased flexibility in domains I and III in pathological samples.

The conducted experiments overall support the obtained conclusions, the manuscript is consistently written and technically sound. However, I would like to give a few recommendations of how the work might be even more improved:

1) Abstract and p.4: "Although the HSA in vitro oxidation does not alter its structure, the alteration of patients' HSA correlates with its redox state." This sentence is a bit confusing. There are many pieces of evidence in the literature, that oxidative stress in vitro can not only alter the HSA /BSA structure, but even cause a formation of the molten globule state (see e.g., [Alessandra Del Giudice, et al. 2016] [Rondeau, Philippe, et al. 2008]). I think this statement requires further elaboration: authors should highlight, that the obtained results contradict several previously published works and give some explanations on that. Moreover, to double-check the result it might be useful to compare the SAXS curves directly, and not only their Fourier transforms (see the next point).

2) p.4: "The identical shape of the $P(r)$ of HSA fractions and HSA control sample validates similar overall conformation in solution." A big effort in this work is done in order to find a statistically significant difference between the HSA shapes under different conditions. Even though the calculation of $P(r)$ functions recently became the routine procedure in SAXS due to the Indirect Fourier Transform (IFT) method originally proposed by [Glatter, 1977], it remains an ill-posed problem requiring certain parametrization. For example, in the GNOM program, there are two adjustable parameters: maximum intraparticle distance D_{max} and the Lagrange regularization parameter α , that are connected to the final shape of $P(r)$ distribution. In other words, given different D_{max} and α , the same SAXS profile can be transformed into a set of different $P(r)$ functions. Therefore, using V_r metric on $P(r)$ functions for validation of similarity of different conformers is somewhat dubious and not straightforward. Instead (or in addition), I would recommend comparing the SAXS profiles directly in reciprocal space using one of the well-established statistical tests, e.g. reduced χ^2 test, CorMap p-value test [Franke et al, 2015], or Anderson-Darling test. This would enable authors to avoid possible misinterpretation due to the inherent ambiguity of the IFT method.

3) Since the model-free analysis is highly focused on determining the flexibility of the HSA domains, I would recommend focusing more on the Kratky plot analysis, rather than on the broadening of $P(r)$ functions (which also can be used, but more as an additional/advanced tool). Kratky plot analysis does not require manipulation with the data and provides direct information on the flexibility of a molecule.

It would be informative to show the normalized Kratky plots (and integrals under them) in addition to D_{max} values against the FA and bilirubin content of HSA as well as MELD score.

4) The authors share their experimental data via Simple SAXS open data repository (<https://saxs-server.herokuapp.com/>), which I believe is a very good practice and facilitates openness and reusability of experimental data. However, the data repository is primarily focused on the data that is not specifically targeted for publication. For data used in publications, there is another dedicated databank available: SASBDB (www.sasbdb.org). Maybe, it would be better to deposit the SAXS data there as well, since it may help to disseminate the obtained results in the wider biophysical community.

We thank the reviewers for their expert and thorough consideration of our manuscript and for posing key questions that have led to an improved presentation of our results and to undertaking new data analysis that further support our conclusions. Both reviewers noted critical new insights into the solution properties of human albumin, along with the general interest by connecting clinical studies with structural biology.

Reviewer #1

Reviewer's opening comments - Paar M. and all propose in this paper a novel approach based on Small Angles X-Rays Scattering to study conformations of HSA coming from patients affected by liver disease. This article constitutes an original approach to correlate clinical observations and a structural data. HSA has been widely studied with analytical methods such as chromatography and mass spectrometry (i.e. Naldi M. Journal of Pharmaceutical and Biomedical Analysis 144(2017) 138-153) to characterize the most abundant modifications affecting HSA structure. The originality of this paper resides in the use of structural method in solution such as SAXS to sense the protein conformation in solution.

In summary, the article is very well written, easy to read and which complements the bibliography on the subject well. However, although the authors had a rigorous approach with the SAXS by using an HPLC online and molecular modeling with normal modes, I have some questions and remarks about the SAXS described in my reports.

Author reply – The authors appreciate the reviewer's comments and thoughtful description of the work. We will attempt to satisfy all concerns the reviewer may have.

Reviewer's comment #1 - the first question concerns the data acquisition. Why the data are truncated at 0.01 Å⁻¹? The data were recorded on a synchrotron beamline and with the setup described in the paper we can expect to obtain a q_{min} value close to 0.004 Å⁻¹ at 2.1m with $\lambda = 1.127 \text{ \AA}$ (i.e. with a laboratory source at $\lambda = 1.54 \text{ \AA}$ and d = 1.2m the q_{min} = 0.005 and q_{max} = 0.5 Å⁻¹). In the paper the most important differences are located in small angles regions which give the difference at high distances in the P(r) representation. Although the q range used by the authors are theoretically sufficient to study a small protein such as HSA, I am surprising to see data cut at 0.01 Å⁻¹.

The Guinier's representations shown in sup. data are good but what happens if you extend the plot to the smallest q values? Did you have some problems with radiation damages or concentration effects (due to the low concentration of salt) which could affect the beginning of the curves? The variations observed in the P(r) plot are minimal and the beginning of the SAXS curves strongly influences the maximal distance of P(r) function.

Author reply #1 –

The Authors appreciate the reviewer's concerns about the truncation of the low q data. In this case the truncation is due to limitations in the beamline setup itself which is being actively addressed. These limitations are currently thought to stem from a combination of the beamstop size and the parasitic scattering from apertures upstream from the sample position. As the lowest-resolution q_{min} value

describes the overall size of the scatterer (R_g), the first data point q_{min} ought to be measured at $q_{min} \leq \pi/D_{max}$. For HSA $D_{max} \sim 90\text{\AA}$ the q_{min} needs to be 0.035\AA^{-1} , thus the $q_{min} = 0.01\text{\AA}^{-1}$ is a very conservative truncation to avoid aberrations in the SAXS profiles. We comment on this procedure in the Methods section.

Please see the following changes on page 14:

“In determining the linear portion of the Guinier region required to obtain an accurate R_g value, the first few points of the curve often require elimination due to limitations in data collection. This leads to a conservative truncation of the low q range to $\sim 0.01\text{\AA}^{-1}$ before continuing analysis.”

Furthermore, we never observed radiation damage in SEC-SAXS mode at the SIBYLS beamline. The HSA elution with the protein concentration $< 1\text{mg/ml}$ flow with the 0.5 ml/min rate into the 1.5 mm thick SEC-SAXS cell and beam size $\sim 0.5\text{ mm}$. Thus, the estimated time that elution is exposed to the X-ray is less than 0.1s and enables radiation damage-free data collection.

Reviewer’s comment #2 - the second point is a remark concerning the figure 1 where are plotted chromatogram. Instead of to show the integrated SAXS signal which is effectively proportional to the concentration, wouldn’t it be more relevant to show the concentration of the protein $[C]$ and in a same time, the value of $I(0)$ divided by the concentration $[C]$? As much as $I(0)$ depends of the concentration and the size of the object, it is a good indicator to check the homogeneity of the elution peak.

Author reply #2 –

The reviewer is correct. The integrated SAXS signal is effectively proportional to the concentration and thus is simply shown to demonstrate the elution of the sample as a function of elution volume and not as quantitative information. We added the UV elution trace to SEC-SAXS chromatogram for selected samples (see new Figure 1). We also show all-samples’ UV elution traces together with determining molecular weights across the main SEC peak (see new Supplemental figure 1). However, our current SEC-SAXS-MALS set up cannot determine protein concentration in the flow-through SAXS sample cell. Dilution in the SAXS sample cell followed by UV and MALS sample cells cause a broadening of the UV chromatogram. Therefore, the SAXS integrative signal and R_g values across the SEC peak are the most accurate indicators of the elution peak's homogeneity. This being said the authors have elected to modify Figure 1 to include additional information.

Please see changes to Figure 1 and Figure 1 caption

Reviewer’s comment #3 - the third point is also a remark and concerns the use of the commercial HSA provided by Sigma. At our laboratory, we use commonly the same HSA provided by this company and we observe always the dimer peak during elution profile in the PBS buffer conditions with a similar column (Bio-Sec3-300 Agilent). You observe only the dimer with the patients P1 and P2 ?

Author reply #3 –

As shown by the SEC-UV chromatograms (see new Supplemental Figure 1), most HSA samples contain some amount HSA dimers. Additionally, we describe in more detail our routine HSA sample preparations that eliminate most of the HSA dimers (see Methods section). In short, filtering the samples through 300kDa cut-off filters may eliminate most of the HSA higher oligomers and further dilute the sample. Variations in dimer amount and relatively low HSA concentration (3-5 mg/ml) may lead to undetectable dimers by SEC-SAXS for some control and patient samples (see new Supplemental Figure 1).

Please see new Supplemental Figure 1 and the following on page 4:

“SEC prior to SAXS separated small amounts of self-associating HSA dimers, which had not been considered in the further analysis (Supplementary Fig. 1a).”

Reviewer’s comment #4 – the fourth point concerns the molecular modelling. The approach combining the comparison with crystallographic structures and all-atom models generated by normal modes is relevant. SAXS is a powerful tool to discriminate structures each other but I'm always reserved when it comes to doing the opposite operation. The approach with normal modes allows to generate possible conformations with a non-aberrant global energy as we often find in modelling algorithms against SAXS data. However, the SAXS stays a low resolution structural method with an important incertitude when models are generated by molecular modeling algorithm. I do not question the results found by the authors, I am convinced by their results thanks to the rigor of their work. But it will be possible to evaluate this incertitude by superimpose 10 (or more) structures just shown in ribbon (just backbone without secondary structures) generated with normal nodes with a chi value close to each other and by highlighting the best model? In this way I suppose we can judge the degree of imprecision of the modeling, the aim being to show the reader that even if the SAXS is a low resolution method it can give relevant indications on structural modifications but with a relative precision that can be easily evaluated. Maybe this could be the subject of an additional figure in the material and method, not in the main text.

Author reply #4 –

We agree with the reviewer that it is important to report uncertainty of our SAXS atomistic modeling. The SAXS base modeling can derive in atomistic models, however they need to be understood as a low-resolution model. The here presented comparison of the atomistic models of patient and control HSA shows HSA conformational variability rather than static structures. As recommended by the reviewer, we show the top-score NMA-SAXS models for two patient and two control HSAs (see new Figure 3c). Although the top-score models have very similar goodness of SAXS fit (χ^2) they vary in the conformation of domain I and III. Larger conformational space of Domain I and III is occupied in patients' HSA

and further validate the larger dynamicity of patient's HSAs (reported in Results).

Please see changes to Figure 3 and Figure 3 caption. Also please see the following changes on page 8:

“The conformational variability between the top-scoring models point out the uncertainty of SAXS based modeling (Fig. 3c). Thus, these atomistic models cannot be understood as unique static structures. Conformational variability of the domain I and III in the top-scoring models of the patient samples is significantly larger than that of control samples (Fig. 3c) and further validate increased dynamicity in patient HSA. We visualized quantitative differences between control and patient conformers by superimposing crystal structure with the best fit model”

Reviewer's final comments – In conclusion, the authors provide a very good and original work combining clinical profiles of patient affected by liver diseases, biochemical measurements, structural information coming from X-ray measurement in solution and molecular modeling. The conclusions are coherent and give a new insight in the study of liver's pathology most specifically on the knowledges of HSA properties.

Author reply – The authors greatly appreciate the reviewer's kind take on our work and hope that we have satisfied their concerns.

Reviewer #2,

Reviewer's opening comments - The manuscript describes the difference in the conformational state of HSA in plasma of healthy people and patients with pathological liver diseases. The correlation between HSA structural changes with some pathological conditions, such as redox state of HSA, redundancy of fatty acids and bilirubin in human plasma, is reported. The authors employ SAXS on real clinical samples, therefore this work not only provides insights into the conformational transitions of the most abundant protein in human blood plasma but also has concrete practical implications. Further, the authors show that the HSA quaternary structure from patients with chronic liver diseases can be used as prognostic markers. To describe the conformational change, the authors utilize the formal SAXS analysis and normal mode analysis in program SREFLEX and find the increased flexibility in domains I and III in pathological samples.

The conducted experiments overall support the obtained conclusions, the manuscript is consistently written and technically sound. However, I would like to give a few recommendations of how the work might be even more improved:

Author reply – The authors appreciate the reviewer's comments and thoughtful description of the work. We will attempt to satisfy all concerns the reviewer may have.

Reviewer's comment #1 – Abstract and p.4: “Although the HSA in vitro oxidation does not alter its structure, the alteration of patients' HSA correlates with its redox state.” This

sentence is a bit confusing. There are many pieces of evidence in the literature, that oxidative stress in vitro can not only alter the HSA /BSA structure, but even cause a formation of the molten globule state (see e.g., [Alessandra Del Giudice, et al. 2016] [Rondeau, Philippe, et al. 2008]). I think this statement requires further elaboration: authors should highlight, that the obtained results contradict several previously published works and give some explanations on that. Moreover, to double-check the result it might be useful to compare the SAXS curves directly, and not only their Fourier transforms (see the next point).

Author reply #1 – We agree with the reviewer that there is some evidence that oxidation has impact on the structure of albumin. However, often (and in the papers mentioned) albumin is oxidized under strenuous conditions with, e.g., hypochlorite (the case in Del Giudice et al. 2016) or by glycooxidation (in the case of Rondeau et al. 2008). We oxidize with cystine which is an extremely mild oxidation and with hydrogen peroxide which is also mild compared to HOCl, with obviously only specific reactions at distinct points. When we oxidized with HOCl (not reported in the paper) under conditions often described in the literature we could not detect any albumin anymore in our HPLC purification approach. On the other hand, even after incubation of albumin with hydrogen peroxide we find distinct chromatographic peaks. This proves that our oxidation conditions are mild compared to those reported frequently.

However, closer investigation of Kratky Plots and minor increase in the Rg values for HMA and HNA1 vs. untreated HSA (see revised Figure 1) suggest subtle conformational variability in between the HSA fractions. The very small peak shift in the Kratky plots of HMA (reduced form on Cys-34) and HNA1 (oxidized form) suggest altered local fold of HSA. The identical broadening of the P(r) function indicates similar overall conformation of HSA fractions as previously stated. This observation is reported in Results section.

Please see changes to Figure 1 and Figure 1 caption. Also please see the following changes on page 4:

“Similar shape and broadening of the P(r) functions indicates similar overall conformation of HSA fractions in solution. However, small peak shift in the Kratky plots (Fig. 1B) and minor increase in the Rg values of HMA and HNA1 (Supplementary Table 2) suggest altered local fold of the reduced and oxidized HSA.”

Reviewer’s comment #2 – p.4: “The identical shape of the P(r) of HSA fractions and HSA control sample validates similar overall conformation in solution.” A big effort in this work is done in order to find a statistically significant difference between the HSA shapes under different conditions. Even though the calculation of P(r) functions recently became the routine procedure in SAXS due to the Indirect Fourier Transform (IFT) method originally proposed by [Glatter, 1977], it remains an ill-posed problem requiring certain parametrization. For example, in the GNOM program, there are two adjustable parameters: maximum intraparticle distance Dmax and the Lagrange regularization

parameter alpha, that are connected to the final shape of $P(r)$ distribution. In other words, given different D_{max} and alpha, the same SAXS profile can be transformed into a set of different $P(r)$ functions. Therefore, using V_r metric on $P(r)$ functions for validation of similarity of different conformers is somewhat dubious and not straight-forward. Instead (or in addition), I would recommend comparing the SAXS profiles directly in reciprocal space using one of the well-established statistical tests, e.g. reduced χ^2 test, CorMap p-value test [Franke et al, 2015], or Anderson-Darling test. This would enable authors to avoid possible misinterpretation due to the inherent ambiguity of the IFT method.

Author reply #2 –

We agree with the reviewer and we have added our validation of similarity using SAXS curves.

Please see Figure 2B and its caption.

We were aware that adjustable parameters, like D_{max} or Lagrange regularization, can transform experimental SAXS data into different $P(r)$ functions. However, the regularization parameters rarely impact the overall $P(r)$ broadening or position of main $P(r)$ peak. Furthermore, the $P(r)$ function is much more sensitive to conformational changes than the individual SAXS parameters or SAXS curves. SAXS curves can also lead to false similarity-comparison due to different signal/noise ratio or imperfect buffer subtraction. The higher signal/noise ratio of C6 and C1 sample may impact the similarity matrix in between the control samples. Whereas the $P(r)$ matrix do not seem to be affected. Never the less the both similarity maps clearly distinguish between control and patient's data sets and correlate with the single SAXS parameters like R_g , area under Kratky plot or D_{max} .

Please see the following changes on page 6-7:

“The heat maps show similarity in between all patients SAXS curves, but also indicate some level of dissimilarities in between control samples (Fig. 2B). We believe that these dissimilarities are related to the different signal to noise ratios in the SAXS curves.”

Reviewer's comment #3 – Since the model-free analysis is highly focused on determining the flexibility of the HSA domains, I would recommend focusing more on the Kratky plot analysis, rather than on the broadening of $P(r)$ functions (which also can be used, but more as an additional/advanced tool). Kratky plot analysis does not require manipulation with the data and provides direct information on the flexibility of a molecule. It would be informative to show the normalized Kratky plots (and integrals under them) in addition to D_{max} values against the FA and bilirubin content of HSA as well as MELD score.

Author reply #3 –

We agree and we appreciate the reviewer's suggestion to use normalized Kratky plots to quantitatively validate conformational flexibility. In our original manuscript we used Kratky plot to show qualitative differences between patients' and controls'

HSA (Figure 3b). We add comparison of normalized Kratky plots between all HSA fractions, HSA controls, and HSA patients (see revised Figure 1). The correlation of R_g values and areas under Kratky plots (see new Figure 1E) clearly show larger dynamicity of patients' HSA. The correlation against FA, bilirubin, and MELD score is reported in Table 2. These analyses further support our original conclusion describing the larger dynamicity of patients' HSAs that correlate to the FA, bilirubin load. Furthermore, only small shift in the Kratky plot of redox fractions HMA, HNA1 suggest small changes in the HSA fold rather than changes in the overall shape. As shown in Table 2 the Kratky plot area correlates significantly to the age of patients + controls. We assume that this is due to the strong correlation of Kratky plot area with HNA1 which we found earlier to increase with age.

Please see changes to Figure 1 and Figure 1 caption. Also see the following changes on pages 4:

“Similar shape and broadening of the $P(r)$ functions indicates similar overall conformation of HSA fractions in solution. However, small peak shift in the Kratky plots (Fig. 1B) and minor increase in the R_g values of HMA and HNA1 (Supplementary Table S2) suggest altered local fold of the reduced and oxidized HSA.”

Also see the following changes on page 6:

“SAXS curves for the monodisperse monomeric state of HSA samples (Supplementary Fig. 2b) were transformed into the normalized Kratky plots (Fig. 1d). The shift in the peak of normalized Kratky plot towards larger $q \cdot R_g$ values indicate conformational change of patient HSAs. By calculating areas under normalized Kratky plots we quantitatively compared these changes. The correlation of these areas with the R_g values determined by the Guinier plot for SAXS curves (Supplementary Fig. 2C) show increased dynamicity of patient HSAs (Fig. 1e, Supplementary Table 2).”

Please see also changes on page 7

In contrast, we found a correlation of the Kratky plot area with age. This may be due to the strong correlation with HNA1 which is increasing with age ¹⁵.

Reviewer's comment #4 – The authors share their experimental data via Simple SAXS open data repository (<https://saxs-server.herokuapp.com/>), which I believe is a very good practice and facilitates openness and reusability of experimental data. However, the data repository is primarily focused on the data that is not specifically targeted for publication. For data used in publications, there is another dedicated databank available: SASBDB (www.sasbdb.org). Maybe, it would be better to deposit the SAXS data there as well, since it may help to disseminate the obtained results in the wider biophysical community.

Author reply #4 – The authors agree with the reviewer on this point and curves are being submitted to SASBDB. The SASBDB IDs are available in Supplementary Table 2.

REVIEWERS' COMMENTS:

Reviewer #1 (Remarks to the Author):

I thank the authors of this article for having taken into consideration the remarks of the referees, thus improving the presentation of the results and giving a relevant article on the subject. As described previously, the article is based on an original approach that combines clinical data and structural approaches.

The care taken in the acquisition of SAXS data allows reliable conclusions to be drawn regarding the link between functional deficiency of HSA and its structural modifications. This study therefore enriches our knowledge of diseases linked to a structural deficiency of HSA. This study also brings a new perspective on the use of SAXS more particularly here in medical studies.

In the attached file you will find my comments on the authors' responses. The authors responded perfectly to my comments and made the changes to their article. The article can therefore be published without further modifications. It is an excellent work.

Reviewer #2 (Remarks to the Author):

I thank the authors for the thoroughly edited revision and believe that all my major concerns were fully addressed. Just a small comment on the direct SAXS profiles comparison: the authors show that all the curves are similar in the (new) fig.2b using the same measure V_r they use for $p(r)$'s. This test (as far as I understood) e.g. does not take the experimental uncertainties (third column) into account. As I wrote earlier, there may be some slight but statistically significant evidence of differences in the profiles if they just try the other metric, e.g. χ^2 , CorMap, or A-D test (those tests are implemented in the new version of program PRIMUS, ATSAS package). Otherwise, I think this is a very interesting and scientifically sound work that deserves to be published in Communications Biology.

Reviewer #2 (Remarks to the Author):

I thank the authors for the thoroughly edited revision and believe that all my major concerns were fully addressed.

Thank you

Just a small comment on the direct SAXS profiles comparison: the authors show that all the curves are similar in the (new) fig.2b using the same measure V_r they use for $p(r)$'s. This test (as far as I understood) e.g. does not take the experimental uncertainties (third column) into account. As I wrote earlier, there may be some slight but statistically significant evidence of differences in the profiles if they just try the other metric, e.g. χ^2 , CorMap, or A-D test (those tests are implemented in the new version of program PRIMUS, ATSAS package). Otherwise, I think this is a very interesting and scientifically sound work that deserves to be published in Communications Biology.

We agree with the reviewer on this point, and we add Figure 2b inset panel to showing a similarity map using the χ^2 metric.

Advantages of V_r over χ^2 can be subtle; however, optimally, V_r highlights key distinctions that are missed by the χ^2 . We developed- V_r -based similarity metrics to distinguished small changes in the fold or conformational variability that primarily impact SAXS in the middle q-region (Hura et al. 2009 Nature Methods). χ^2 is often used to provide binary “accept or reject” criteria for judging whether a proposed atomic model is consistent with experimental SAXS data. We optimized V_r not only to distinguish when structures are the same or different but also to show the degree of similarity (Hura et al. 2009 Nature Methods, <https://bl1231.als.lbl.gov/saxs-similarity/>).

Additionally, we also tried to applied χ^2 and CorMap metrics implemented in the ATSAS package; unfortunately, the error in binning of SAXS profiles prevents us from using this platform.

Hura, G. L. *et al.* Robust, high-throughput solution structural analyses by small angle X-ray scattering (SAXS). *Nature methods* **6**, 606-612 (2009).